# Anticholinesterase and Serotoninergic Evaluation of Benzimidazole–Carboxamides as Potential Multifunctional Agents for the Treatment of Alzheimer’s Disease

**DOI:** 10.3390/pharmaceutics15082159

**Published:** 2023-08-19

**Authors:** Daria A. Belinskaia, Polina A. Voronina, Denis V. Krivorotov, Richard O. Jenkins, Nikolay V. Goncharov

**Affiliations:** 1Sechenov Institute of Evolutionary Physiology and Biochemistry, Russian Academy of Sciences, Thorez 44, St. Petersburg 194223, Russia; 2Research Institute of Hygiene, Occupational Pathology and Human Ecology, Federal Medical Biological Agency, p.o. Kuzmolovsky, St. Petersburg 188663, Russia; 3Leicester School of Allied Health Sciences, De Montfort University, The Gateway, Leicester LE1 9BH, UK

**Keywords:** Alzheimer’s disease, benzimidazolones, cholinesterases, serotonin receptors, biochemical analysis, molecular docking, molecular dynamics

## Abstract

The etiology and pathogenesis of Alzheimer’s disease are multifactorial, so one of the treatment strategies is the development of the drugs that affect several targets associated with the pathogenesis of the disease. Within this roadmap, we investigated the interaction of several substituted 1,3-dihydro-2-oxo-1*H*-benzimidazol-2-ones with their potential molecular targets: cholinesterases (ChE) and three types of the G_s_-protein-coupled serotonin receptors (5-HTR) 5-HT_6_, 5-HT_4_ and 5-HT_7_ (5-HT_4_R, 5-HT_6_R and 5-HT_7_R, respectively). A microplate modification of the Ellman method was used for the biochemical analysis of the inhibitory ability of the drugs towards ChE. Molecular modeling methods, such as molecular docking and molecular dynamics (MD) simulation in water and the lipid bilayer, were used to study the interaction of the compounds with ChE and 5-HTR. In vitro experiments showed that the tested compounds had moderate anticholinesterase activity. With the help of molecular modeling methods, the mechanism of interaction of the tested compounds with ChE was investigated, the binding sites were described and the structural features of the drugs that determine the strength of their anticholinesterase activity were revealed. Primary in silico evaluation showed that benzimidazole–carboxamides effectively bind to 5-HT_4_R and 5-HT_7_R. The pool of the obtained data allows us to choose N-[2-(diethylamino)ethyl]-2-oxo-3-(tert-butyl)-2,3-dihydro-1*H*-benzimidazole-1-carboxamide hydrochloride (compound **13**) as the most promising for further experimental development.

## 1. Introduction

From 1990 to 2019, the incidence and prevalence of Alzheimer’s disease (AD) and other dementias increased by 147.95% and 160.84%, respectively [1]. Commercially available AD drugs donepezil, rivastigmine, galantamine (ChE inhibitors) and memantine (NMDA-receptor antagonist) can improve cognitive and behavioral parameters, but their clinical efficacy remains modest [2]. Therefore, the development of an effective and safe pharmacotherapy for AD is still an urgent task.

There are several hypotheses regarding the causes of AD (discussed in detail in some recent reviews [3,4,5]), including the amyloid cascade (the main and most studied), tau propagation, neuroinflammation, oxidative stress, mitochondrial dysfunction and cholinergic and vascular hypotheses.

Regardless of the reasons of the initiating and development of AD, the cholinergic system is essentially damaged in AD patients; therefore, therapeutic approaches aiming at restoring the level of acetylcholine are widespread in clinical practice [5,6]. However, since the etiology and pathogenesis of neurodegenerative diseases—including AD—are multifactorial, the development of the drugs that can affect several targets associated with the pathogenesis of the disease is a reasonable strategy. Multitarget drugs have a number of advantages: more predictable pharmacokinetics, a lower probability of drug–drug interactions, minimized drug resistance and maximized patient compliance [7]. For example, in order to expand the functional properties of the known AD drugs, N-alkylated tacrine derivatives have been obtained [8], as well as hybrid molecules of tacrine with tryptophan, coumarin, 7-chloroquinoline, indole or benzothiasoles [9,10,11,12]. Hybrid structures have been synthesized that combine certain pharmacophores, for example, γ-carbolines, carbazoles, phenothiazines and aminoadamantanes [13,14,15,16], amiridine and piperazine [17], pyrazolopyridine and tetrahydroacridine [18], tacrine–dipicolylamine dimers [19], chrysin and rivastigmine [20] and naringenin derivatives [21].

The hybrid molecules that combine anticholinesterase activity and high affinity for G-protein-coupled receptors (GPCRs) seem to be promising. Thus, ChE inhibitors have been synthesized that selectively affect histamine H3 receptors (hH3R, regulates the release of acetylcholine, ACh) [22] and cannabinoid type 2 receptors (hCB2R, modulate the processes of the “silent” period of AD including neuroinflammation, oxidative stress and mitochondrial dysfunction) [23,24]. G_s_-protein-coupled serotonin receptors (5-HTR) 5-HT_6_, 5-HT_4_ and 5-HT_7_ (5-HT_4_R, 5-HT_6_R and 5-HT_7_R, respectively) are of particular interest as molecular targets for AD management. Thus, the stimulation of 5-HT_4_R slows down the formation of extracellular amyloid plaques [25,26]. The 5-HT_4_R agonist prucalopride was shown to enhance cognitive function in healthy participants, bilaterally increasing hippocampal activity and activity in the right angular gyrus [27,28]. The 5-HT_4_R partial agonist usmarapride was demonstrated to produce elevation in cortical acetylcholine levels in rats and to increase levels of soluble amyloid precursor protein alpha, a potential mechanism to reverse toxic Aβ peptide pathology, which makes it a promising drug for alleviating the cognitive dysfunction in AD patients [29].

5-HT_6_R are localized exclusively in the central nervous system, where they modulate the levels of γ-aminobutyric acid (GABA) and glutamate, thereby promoting the secondary release of dopamine, norepinephrine, acetylcholine and other neurotransmitters [30,31]. A number of preclinical tests revealed that both agonists and antagonists of 5-HT_6_R can improve cognition capacity; however, clinical trials showed the weak effect of 5-HT_6_R modulators for AD management [32,33]. It has been suggested that 5-HT_6_R agents could reveal their full potential in the formulation of multitargeting drugs [32].

Recent experimental data suggest that 5-HT_7_R (involved in learning and memory processes) may also be a suitable therapeutic target for the management of neurodegenerative disorders [34]. LP-211, specific agonists of 5-HT_7_R, were shown to revert neuronal damage and cognitive impairment induced by Aβ in a neurodegenerative animal model [35]. The 5-HT_7_R stimulation with AS19 (another selective agonist) improved synaptic dysfunction via the reduction of apoptosis in the rat hippocampus, which makes it a promising drug for preventing the progression of AD [36].

In our previous work, we synthesized a number of substituted 1,3-dihydro-2-oxo-1*H*-benzimidazol-2-ones, potentially combining the functions of ChE inhibitors and agonists of the G_s_-protein-coupled 5-HTR, and evaluated their toxicity in silico [37]. The purpose of the present study is to investigate the anticholinesterase activity of the synthesized benzimidazole–carboxamides and their ability to bind to three types of the G_s_-protein-coupled 5-HTR (5-HT_4_R, 5-HT_6_R and 5-HT_7_R) using biochemical and molecular modeling methods.

## 2. Materials and Methods

### 2.1. Choice and Synthesis of the Benzimidazole–Carboxamides

In our previous work [37], we performed the theoretical calculation of some physicochemical and toxicological parameters of compounds from a group of substituted 1,3-dihydro-2-oxo-1*H*-benzimidazol-2-ones (the general formula is shown in Figure 1). The following parameters were calculated: the distribution coefficient logP, which characterizes the ratio of the solubility of a compound in octanol and water; the ability to pass through the blood–brain barrier (BBB) logBB, based on the distribution of substances between blood plasma and the brain parenchyma; and the semi-lethal dose LD_50_ (mg/kg) for mice and rats with different routes of administration.

Based on the data obtained, the most promising compounds were selected, which were subsequently synthesized. A convenient preparative method for the synthesis of 1,3-dihydro-2*H*-benzimidazol-2-ones based on the nucleophilic substitution reaction in *o*-dinitrobenzene was developed at the Research Institute of Hygiene, Occupational Pathology and Human Ecology. The synthesis procedure was briefly described in our previous work [37]. Here, for the first time, we publish in detail all the stages of the synthesis of the intermediate and target compounds; the data are presented in the Appendix A. The yields of the obtained compounds were determined with the ratio of the actual output to the theoretical one. The purity and authenticity of the obtained compounds were confirmed with nuclear magnetic resonance.

For the subsequent biochemical analysis of the inhibitory activity of benzimidazole–carboxamides towards AChE and BChE, eight low-toxic compounds were selected that differ in R2 or R3 substituents; the size of the substituents; the presence or absence of a fluoride atom in their structure; and the capability of additional binding in vivo. The structures of the selected compounds are shown in Figure 1.

The substituents of most of these compounds are slight variations of the substituents of BIMU-8 (compound **15**), which has been known for over 30 years as a selective agonist of 5-HT_4_R [38]. The structure of substituent R3 of compound **24** is typical for a number of antihistamines (cyclizine, cetirizine, hydroxyzine, etc.).

### 2.2. Anticholinesterase Activity of Benzimidazole–Carboxamides

The biochemical analysis of the inhibitory ability of the drugs against AChE and BChE was performed using a microplate modification of the Ellman method [39]. The following reagents were used: 5,5′-dithio-bis-2-nitrobenzoic acid (DTNB, Sigma-Aldrich, St. Louis, MO, USA), acetylthiocholine (ATCh, Sigma-Aldrich), butyrylthiocholine (BTCh, Sigma-Aldrich), human erythrocyte AChE (Biomed, Auckland, New Zealand), human plasma BChE (Mechnikov Scientific Research Institute of Vaccines and Serums, St. Petersburg, Russia), phosphate-buffered saline (PBS; pH 7.2; Biolot, Ankara, Türkiye) and distilled water (pH 5.97, Biolot).

AChE and BChE were dissolved in PBS to obtain solutions of 1 mg/mL. Due to its high activity, human BChE was used at a 1:30 dilution. Solutions of the studied compounds were prepared with serial dilutions; their concentration in the reaction mixture was in a range from 0 to 300 μM. Double-distilled water, ethanol and dimethyl sulfoxide (DMSO, up to 15% of the total solvent volume) were used as solvents. When using DMSO or ethanol, an additional control was performed to assess the effect of the solvent itself. Protein (AChE of BChE) was incubated for 5 min at 37 °C and then the coloring solution was added, which contained a 2 mM freshly prepared DTNB solution and the substrate (ATCh/BTCh) in the concentration range of 200–800 μM. The kinetics were recorded for 15 min at 37 °C at a wavelength of 405 nm. Activity (μM × min^−1^) was calculated using the molar extinction coefficient 0.01415 μmol^−1^ × cm^−1^: (V/ε × l × v) × (ΔE/Δt), where V is the volume of the analyzed mixture (μL); v is the volume of the enzyme solution (μL); ΔE/Δt is the change in extinction in 1 min; l is the thickness of the absorbing layer (cm); and ε is the coefficient of molar extinction of 2-nitro-5-sulphonyl benzoic acid. Each measurement was repeated three times.

The type of inhibition was evaluated using Lineweaver–Burk plots. The dissociation constant of the enzyme-inhibitor complex (K_i_), the maximum rate (V_max_) and thermodynamic cooperativity factor α were calculated using the Graphpad Prism 8.0 program (GraphPad Software Inc., San Diego, CA, USA) according to the model “enzymatic kinetics-inhibition—type of inhibition”.

### 2.3. Building of 3D Models for Molecular Modeling

Ammonium groups of compounds **13**, **24** and BIMU-8 were protonated and positively charged. Three-dimensional models of protonated drugs were built and optimized with the energy minimization method using HyperChem 8.0 software (Hypercube Inc., Gainesville, FL, USA) [40]. The three-dimensional models of human ChE and 5-HTR were downloaded from the protein data bank (PDB) [41]. The following structures were used: 4ey7 (chain A) for AChE [42]; 3djy for BChE [43]; and 7xta (chain R), 7xtb (chain R) and 7xtc (chain R) for 5-HT_4_R, 5-HT_6_R and 5-HT_7_R, respectively [44]. Water molecules and ligands were removed from the downloaded structures, missing atoms were added using the Visual Molecular Dynamics v.1.9.4a53 software package (VMD, University of Illinois Urbana-Champaign, Champaign, IL, USA) [45] and mutated residues were converted into the native ones using Discovery Studio Visualizer v21.1.0.020298 (BIOVIA, San Diego, CA, USA) [46]. Available 3D structures of 5-HT_4_R, 5-HT_6_R and 5-HT_7_R contained missing regions that were not resolved with electron microscopy. The following amino acids were missing: 1–17, 226–253 and 329–387 in 5-HT_4_R; 1–25, 228–262 and 339–440 in 5-HT_6_R; and 1–79, 280–323, 355–359 and 404–432 in 5-HT_7_R. All these regions do not overlap with serotonin-binding sites. In our computational experiments, we did not rebuild these gaps, since we were more interested in the binding modes of the drugs within the receptors, and not in the conformational changes of 5-HTR complexes with G-proteins caused by ligand binding.

### 2.4. Molecular Docking

The search for drug-binding sites was performed with the method of blind docking (the search area was the entire surface of the protein) using SwissDock online service (Swiss Institute of Bioinformatics in Lausanne, Switzerland) [47]; the protein molecule was set to be rigid, and the conformation of the ligand could vary. The targeted molecular docking of the ligands into the found binding sites of ChE and 5-HTR was performed using Autodock Vina v.1.1.2 software (The Scripps Research Institute, La Jolla, CA, USA) [48]. A search area, 25 × 25 × 25 Å^3^, was set in the studied protein binding site. The parameter “exhaustiveness” (determining the number of runs and the amount of computational effort) was set to 20. The parameter “energy_range” (maximum energy difference between the best binding mode and the worst one) was set to 3 kcal/mol. The number of the most optimal conformations in the output file (num_modes) was set to 10. The conformation of the ligands could vary, and the protein remained rigid. The most energetically favorable conformation was selected for the further analysis.

### 2.5. Molecular Dynamics of Drug-ChE Complexes in Water

The ligand-ChE complexes were placed virtually into a cubic periodic box filled with water molecules. The TIP3P water model (transferable intermolecular potential with 3 points) was used to describe water molecules [49]. To neutralize a system, sodium or chloride ions were added. Conformational changes of the complexes were calculated with 100 ns molecular dynamics (MD) simulation using GROMACS2019.4 software (University of Groningen, the Netherlands) [50] and a CHARMM27 force field [51]. Temperature (300 K) and pressure (1 bar, isotropic) were kept constant using a V-rescale thermostat [52] and Parrinello–Rahman barostat [53], with coupling constants of 0.1 ps and 2.0 ps, respectively. Long-range electrostatic interactions were treated with the particle-mesh Ewald method [54]. Lennard-Jones interactions were calculated with a cutoff of 1.0 nm. The LINCS algorithm (linear constraint solver for molecular simulations) was used to constrain bond length [55]. Before running the MD simulations, all the structures were minimized with steepest-descent energy minimization and equilibrated under NVT (1000 ps) and NPT (5000 ps) ensembles. The time step for MD simulation was 0.002 ps. The analysis of the trajectories was performed using the VMD v.1.9.4a53 software package (University of Illinois Urbana-Champaign, Champaign, IL, USA) [45].

### 2.6. Molecular Dynamics of the Drug Complexes with 5-HT_4_R in Lipid Bilayer

Using the online service CHARMM-GUI Membrane Builder (Lehigh University, Bethlehem, PA, USA) [56], the complexes of ligands with 5-HT_4_R obtained with molecular docking were virtually inserted into an 80 × 80 Å lipid bilayer consisting of 1-palmitoyl-2-oleoyl-sn-glycero-3-phosphocholine (POPC) molecules. The total number of POPC molecules was 151 (76 in the upper lipid layer and 75 in the lower one). A solvent (water) was virtually added to the system; the thickness of the water layer above and below the membrane was set as equal to 15 Å. Sodium and chloride ions (NaCl) were added to the system to neutralize the charge; the concentration of NaCl was set as equal to 0.15 M. Conformational changes of the complexes were calculated with 100 ns molecular dynamics (MD) simulation using GROMACS2019.4 software [50] and a CHARMM36m force field [57]. First, the system, constructed using the CHARMM-GUI Membrane Builder, was optimized with the energy minimization method and further equilibration with a gradual impairment of the restraints imposed on the movement of atoms. The total duration of six equilibration stages was 39 ns. Then, the conformational changes of the system were simulated using 100 ns MD without restraints with an integration step of 0.002 ps. Temperature (303.15 K) and pressure (1 bar, semi-isotropic) were kept constant using a Nose–Hoover thermostat [58] and Parrinello–Rahman barostat [53], with coupling constants of 1.0 ps and 5.0 ps, respectively. Long-range electrostatic interactions were treated with the particle-mesh Ewald method [54]. Lennard-Jones interactions were calculated with a cutoff of 1.2 nm. The LINCS algorithm was used to constrain bond length [55]. The analysis of the trajectories was performed using the VMD v.1.9.4a53 software package (University of Illinois Urbana-Champaign, Champaign, IL, USA) [45].

## 3. Results

### 3.1. Anticholinesterase Activity of Benzimidazole–Carboxamides In Vitro

The results of measuring the anticholinesterase activity of benzimidazole–carboxamides were briefly reported in our previous work [37], and, here, we present the extended analysis of the data. The results of the analysis of the inhibitory ability of the selected compounds towards AChE are presented in Table 1 and shown in Appendix A. The obtained experimental values of the inhibition constant for compounds **15** and **16** indicate that introducing a trifluoromethyl group enhances the inhibitory activity but does not change the type of inhibition (mixed type both for compounds **15** and **16** with comparable values of α; Table 1 and Appendix A). For compounds **22** and **23**, as well as for compounds **15** and **16**, the trifluoromethyl group in the R1 substituent enhances the inhibitory effect; however, in the case of **22** and **23**, the addition of the trifluoromethyl group changes the type of inhibition from competitive to mixed (Table 1 and Appendix A). The replacement of the isopropyl group by the tert-butyl one in substituent R2 (compound **22** vs. **13**) enhances the inhibitory effect, but does not affect the type of inhibition (both compounds **22** and **13** are competitive inhibitors; Table 1 and Appendix A). The replacement of isopropyl substituent R2 with the cyclopropyl one (compound **22** vs. **27** and **28**) decreases the inhibitory activity against AChE and changes the type of inhibition: competitive for compound **22** (Table 1 and Appendix A) and mixed for compounds **27** and **28** with comparable values of α (Table 1 and Appendix A). The replacement of the diethyl group in substituent R3 (compound **27**) with the dimethyl group (compound **28**) affects neither the inhibitory activity nor the type of inhibition. However, a more radical replacement of substituent R3 with the [(4-chlorophenyl)(phenyl)methyl]-piperazin-1-yl group significantly enhances the inhibitory ability: compound **24** is by far the strongest inhibitor of AChE, compared to the other compounds tested (Table 1 and Appendix A). Compound **24** is the only strictly non-competitive AChE inhibitor among all studied compounds.

The results of the analysis of the inhibitory ability of the selected compounds against BChE are presented in Table 2 and Appendix A. Due to its high activity, human BChE was used at a 1:30 dilution. The obtained experimental values of the inhibition constant for compounds **15** and **16** indicate that introducing a trifluoromethyl group weakens the inhibitory activity but does not change the type of inhibition (competitive type for both compounds **15** and **16**; Table 2 and Appendix A). In contrast to the pair of compounds **15** and **16**, for compounds **22** and **23**, the trifluoromethyl group in substituent R1 enhances the inhibitory effect and changes the type of inhibition from competitive to non-competitive (Table 2 and Appendix A). The replacement of the isopropyl group with the tert-butyl group in substituent R2 (compound **22** vs. **13**) enhances the inhibitory effect and changes the type of inhibition from competitive (Table 2 and Appendix A) to non-competitive (Table 2 and Appendix A). The replacement of isopropyl substituent R2 with the cyclopropyl one (compound **22** vs. **27**) affects neither the inhibitory activity against BChE nor the type of inhibition, while the replacement of the diethyl group in substituent R3 (compound **27**) with the dimethyl group (compound **28**) weakens the inhibitory effect and changes the type of inhibition from competitive to non-competitive (Table 2 and Appendix A). Compound **24**, which contains the [(4-chlorophenyl)(phenyl)methyl]-piperazin-1-yl group in the R3 substituent, has no inhibitory effect on BChE.

Thus, according to the data of the biochemical experiment, compound **24** is the best inhibitor of AChE, and compound **13** is the best inhibitor of BChE. Next, we studied the interaction of these compounds with potential targets (ChE and 5HT_4_R/5HT_6_R/5HT_7_R) using molecular modeling methods.

### 3.2. Search for Binding Sites on the Surface of AChE and BChE

Using compound **24** (the best inhibitor for AChE) as an example, we searched for possible binding sites for benzimidazolones on the surface of AChE. For this purpose, we performed the blind molecular docking of compound **24** into the AChE molecule, setting the entire protein surface as the search area. The result of the docking is shown in Figure 2.

The active site gorge connecting the catalytic active site (CAS) and the peripheral anionic site (PAS, forms the entrance to the active site) of AChE is the binding site for competitive inhibitors (Site S1a). Depending on the structure, competitive inhibitors can either bind to CAS, or, as in the case of donepezil [59], occupy the entire space of Site S1a between PAS and CAS. Interacting with S1a, competitive inhibitors prevent native substrate acetylcholine (ACh) from entering the active site. None of the found possible positions of compound **24** on the surface of AChE coincide with Site S1a; therefore, according to the blind docking data, compound **24** cannot bind to Site S1a of AChE.

As can be seen from Figure 2, there are two major clusters of the conformations of compound **24** on the AChE surface. We designated the places of their localization as sites S2a and S3a. These found binding sites topologically do not coincide with CAS; that is, they are allosteric centers. By interacting with these sites, a drug can inhibit enzyme activity non-competitively via allosteric modulation. We neglected several minor clusters (Figure 2) and did not consider them as independent sites for compound **24**.

In Site S2a, the following amino acids are involved in the binding of compound **24**: Pro232, Asn233, Gly234, Pro235, Trp236, Thr238, Val239, Gly240, Met241, Glu243, Arg247, Arg296, Val303, Thr311, Pro312, Glu313, Ala314, Asn317, Val367, Pro 368, Val370, Ser371, Asp404, His405, Cys409, Pro410, Gln413, Trp532, Asn533, Leu536, Pro537, Leu540 and Ser541. In Site S3a, amino acids Gly79, Phe80, Glu81, Gly82, Met85, Asp131, Val132, Ala434, Ser435, Thr436, Leu437, Set438, Trp439, Tyr449, Glu452, Ile457, Ser462, Arg463, Asn464 and Tyr465 participate in the binding of the drug.

Previously, two allosteric sites on the AChE surface were described [60]. According to Roca et al. [60], the first allosteric center includes amino acids Pro232, Asn233, Gly234, Pro235, Trp236, Thr238, Val239, Gly240, Glu243, Arg246, Arg247, Leu289, Por290, Gln291, Ser293, Arg296, Phe 297, Val300, Thr311, Pro312, Glu313, Pro368, Gln369, Val370, Asp404, His405, Cys409, Pro410, Gln413, Trp532, Asn533, Leu536, Pro537 and Leu540. The second one includes amino acids Glu81, Gly82, Glu84, Met85, Asn87, Asn89, Leu130, Asp131, Val132, Thr436, Leu437, Ser438, Trp439, Tyr449, Glu452, Ile457, Ser462, Arg463, Asn464 and Tyr465. Thus, the binding sites found by us for compound **24** coincide with the sites of AChE allosteric modulation described in the literature.

Next, using compound **13** (the best inhibitor for BChE) as an example, we searched for possible benzimidazolone binding sites on the surface of BChE. For this purpose, similarly to AChE, we performed the blind docking of compound **13** into the BChE molecule, setting the entire protein surface as the search area. The result of the docking is shown in Figure 3.

According to the obtained data, compound **13** can bind at two sites. The first of them is Site S1b, which, as in the case of AChE, is the cavity connecting CAS and PAS. Compound **13** can also bind at Site S2b in the vicinity of amino acids Pro230, Val233, Thr234, Ser235, Glu238, Arg242, Val280, Tyr282, Gly283, Thr284, Leu286, Ser287, Val288, Phe357, Phe358, Pro359, Val361 and Tyr396. When AChE and BChE molecules were superimposed, it could be seen that Site S2a on the surface of AChE and Site S2b on the surface of BChE partially overlapped (Site S2b is part of Site S2a, but smaller). As in the case of AChE, we neglected the minor cluster of conformations of compound **13** on the surface of BChE (Figure 3) and did not consider it as an independent site for this drug.

The allosteric site of BChE found by us has been described earlier. Thus, in [61], the interaction of alkaloids with ChE was studied and, according to molecular modeling data, 7-epi-javaniside binds on the surface of BChE surrounded by amino acids Glu238, Arg242, Thr284, Pro359 and Pro281. According to molecular modeling data, one of the aminoguanidinium salts of carboxylic acids studied in [62] also binds on the surface of BChE in the vicinity of Val233, Glu238, Arg242 and Val288.

### 3.3. Docking of Benzimidazolones into the Binding Sites of AChE

At the next stage, we performed the targeted molecular docking of compounds **13** and **24** into all found binding sites: S1a, S2a and S3a of AChE and S1b and S2b of BChE. Additionally, the ACh molecule was docked to the active sites of AChE and BChE (sites S1a and S1b) as the control experiment. Estimated values of binding free energies (ΔG) are presented in Table 3.

The most energetically favorable conformations of compound **13** within sites S1a, S2a and S3a of AChE are shown in Figure 4A, Figure 4B and Figure 4C, respectively. The values of free binding energy for these conformations are −7.5, −6.7 and −5.7 kcal/mol, respectively (Table 3).

In Site S1a, compound **13** occupies the space between CAS and PAS of AChE (Figure 4A, left panel), while the benzimidazole ring of the ligand forms pi–pi interaction with Trp286 and Tyr341, and the amino group of the ligand forms pi–cationic interaction with Tyr337. We aligned the obtained pose of compound **13** in Site S1a with the three-dimensional structure of the complex of AChE with a reversible ChE inhibitor, donepezil, obtained with an X-ray analysis and published in the PDB database [59]. The result is presented in the right panel of Figure 4A. The benzimidazole ring of donepezil also forms pi–pi interaction with Trp286 and Tyr341, while its amino group forms pi–cationic interaction with Tyr337. This agreement between computational and experimental data validates the approach that we use.

According to molecular docking data, in allosteric Site S2a, the hydrogen atom of the amino group of the drug interacts with the oxygen atom of the hydroxyl group of Thr238. The benzimidazole ring and its substituents bind in the vicinity of amino acids Pro235, His405, Leu536, Pro537, Leu540 and Trp532 (Figure 4B). In Site S3a, the oxygen atom of the amide group of compound **13** interacts with the hydrogen atom of the sidechain of Asn464, and the hydrogen atom of the amino group of the ligand interacts with the oxygen atom of the backbone of Thr436. The benzimidazole ring and its substituents bind in the vicinity of amino acids Glu81, Arg463, Tyr465 and Leu467 (Figure 4C).

According to the calculated values of free energy (Table 3), compound **13** most effectively, among all AChE sites, interacts with the site of competitive inhibitors S1a. Compound **13** can also interact with the allosteric sites S2a and S3a. We checked the strength of this binding with MD simulation (see Section 3.5).

Although blind docking showed that compound **24** did not interact with Site S1a, we docked compound **24** to this site as the control experiment. According to the obtained data, the values of binding free energy are positive for all the found conformations. Positive values of ΔG indicate that compound **24** cannot bind in Site S1a, which is consistent with the data of blind docking described above, as well as with experimental data indicating that compound **24** is a non-competitive AChE inhibitor (Appendix A). The most energetically favorable conformations of compound **24** in sites S2a and S3a are shown in Figure 5A and Figure 5B, respectively. The free binding energies ΔG for these conformations are −7.9 kcal/mol and −7.4 kcal/mol (Table 3), respectively. In Site S2a (Figure 5A), the benzimidazole ring of the drug binds in the vicinity of aliphatic amino acids Val303 and Gly240 and the CH_3_ group of Thr311. The benzene rings of the bicyclic group are bound in the vicinity of sidechains of the aliphatic amino acids Leu536, Pro537 and Leu540. According to the geometry of the obtained complex, the chlorine atom of the ligand forms a halogen bond with the backbone oxygen atom of Pro537. In the case of Site S3a, compound **24** binds near amino acids Glu81, Thr 436, Leu437, Arg463, Asn464 and Tyr465. According to the geometry of the complex, no strong electrostatic interactions (salt bridges and hydrogen or halogen bonds) are formed.

According to the calculated values of ΔG, Site S2a is the preferred site for the interaction of compound **24** with AChE. However, since the ΔG values for sites S2a and S3a are quite close, at the next stage, we simulated the conformational changes of the obtained complexes with the MD method; the results are described below.

The estimated ΔG for the ACh-AChE complex is −4.7 kcal/mol, so compounds **13** and **24** bind with AChE more efficiently than the natural substrate.

### 3.4. Docking of Benzimidazolones into the Binding Sites of BChE

The most energetically favorable conformations of compound **13** in sites S1b and S2b of BChE are shown in Figure 6A and Figure 6B, respectively. The values of free binding energy for these conformations are −6.8 and −6.6 kcal/mol, respectively (Table 3).

In Site S1b, compound **13** occupies the space between CAS and PAS of BChE, while the benzimidazole ring of the ligand forms pi–pi interaction with Tyr332, and the amino group of the drug forms a salt bridge with Glu197 and pi–cationic interaction with Trp82 (Figure 6A). In Site S2b, the benzimidazole ring of the drug and its substituents bind near amino acids Phe358, Pro359, Asn397, Tyr396, Leu286 and Ser287; the benzimidazole ring of compound **13** forms Y-shaped pi–pi interaction with Phe358. The amino group of the drug binds near Arg242, forming a p-cationic bond (Figure 6B).

According to the calculated values of ΔG (Table 3), compound **13** interacts with the site of competitive inhibitors S1b and with the allosteric Site S2b with approximately equal efficiency. We subsequently calculated the strength of this binding with MD simulation (see Section 3.5). The estimated value of free binding energy for the ACh-BChE complex is −4.3 kcal/mol (Table 3), i.e., compound **13** interacts with BChE more efficiently than the natural substrate. According to the calculated values of ΔG (Table 3), compound **24** interacts with Site S1b of BChE with practically zero efficiency (ΔG = −0.3 kcal/mol) (Table 3). Thus, compound **24** does not compete with ACh for binding to the active site. In the case of Site S2b, the ΔG values for the **24**-BChE complex were positive, indicating no interaction.

### 3.5. Interaction of Benzimidazolones with AChE and BChE According to MD Simulation

Using the method of MD, we modelled the conformational changes of the complexes of compounds **13** and **24** with the sites of their interaction with ChE obtained with molecular docking. These complexes are compound **13** with all sites of AChE and BChE, and compound **24** with Sites S2a and S3a of AChE.

The movement of compound **13** during the simulation in three AChE ligand-binding sites is shown in Figure 7. According to the data obtained, compound **13** remains bound in Site S1a; however, during the simulation, the drug gradually moves towards the exit from the site (Figure 7A). In the case of Site S2a, the ligand remains in the site, but its position within is not rigidly fixed, and the drug molecule moves freely around the site (Figure 7B). In the case of Site S3a, compound **13** leaves the site and, after 100 ns of simulation, binds to PAS, which is the entrance to the site for competitive inhibitors S1a (Figure 7C). This result additionally confirms our in vitro data that compound **13** is a competitive inhibitor of AChE. Thus, according to both docking data (Table 3) and MD data, compound **13** interacts most strongly with Site S1a.

The movement of compound **24** in two allosteric ligand-binding sites of AChE during the simulation is shown in Figure 8. During 100 ns of simulation, compound **24** remains in Site S2a, although its benzimidazole group gradually moves to another part of the site (Figure 8A). In the final stage of the simulation, it binds in the environment of amino acids Gln369 and Val370, while its phenyl groups remain close to amino acids Leu536, Pro537 and Leu540. In the case of Site S3a, the drug almost immediately leaves the site and remains in an unbound state during the entire simulation (Figure 8B). It is interesting to note that compound **13**, after leaving Site S3a, binds with the entrance to the site of competitive inhibitors S1a, and compound **24** does not interact with this region of AChE. This agrees with the experimental data that compound **13** is a competitive AChE inhibitor, and compound **24** is not. Thus, according to the obtained data, Site S2a is the only site of interaction of compound **24** with AChE.

The movement of compound **13** in two BChE ligand-binding sites during the simulation is shown in Figure 9. In the case of Site S1b, compound **13** remains inside the site (Figure 9A). During the simulation, the benzimidazole group sinks a little deeper into the site due to rotation around the immobile cationic group, which keeps pi–cationic interaction with Trp82. In the case of Site S2b, during the first 10 ns of the simulation, the drug molecule moves to another region on the BChE surface (S2b′) and remains there for the remaining 90 ns of the simulation, surrounded by amino acids Cys400, Pro401, Glu404, Trp522, Thr523 and Phe526 (Figure 9B). It is interesting to note that, when the primary sequences of human AChE and BChE are aligned, residues Cys400, Pro401, Glu404, Trp522, Thr523 and Phe526 of BChE correspond to residues Cys409, Pro410, Gln413, Trp532, Asn533 and Leu536 of AChE, which are a part of allosteric Site S2a. That is, the allosteric site on the surface of BChE S2b′ identified with MD simulation corresponds to allosteric Site S2a of AChE. Thus, according to molecular docking (Table 3) and MD simulation (Figure 9), compound **13**, when interacting with BChE, can bind both to the site of competitive inhibitors S1b and to the allosteric site(s) (S2b and/or S2b′).

### 3.6. Interaction of Compounds ***13***, ***15*** and ***24*** with 5-HT_4_R, 5-HT_6_R and 5-HT_7_R According to Molecular Docking Data

At the next stage, with the help of molecular modeling methods, we performed an initial evaluation of the effectiveness of the binding of compounds **13** (the best BChE inhibitor), **24** (the best AChE inhibitor) and **15** (BIMU-8, a selective agonist of 5-HT_4_R) with 5-HT_4_R, 5-HT_6_R and 5-HT_7_R. The results of molecular docking of the compounds into the binding sites of the studied 5-HTR are presented in Table 4 and Figure 10.

According to molecular docking data, compounds **13**, **24** and BIMU-8 bind in the same cavity inside 5-HT_4_R as serotonin does. However, due to their larger steric size, these drugs bind not at the bottom of this cavity (like serotonin) but closer to its entrance (Figure 10A). For verification, we performed the molecular docking of the drugs into the serotonin-binding site of 5-HT_4_R, reducing the size of the search area. We were able to find the positions of compound **13** and BIMU-8, which were located in the same place as serotonin (at the bottom of the pocket). However, ΔG values for these conformations are −6.4 kcal/mol and −3.5 kcal/mol for compound **13** and BIMU-8, respectively, which are higher than the ΔG values for the “upper” positions of the drugs. This means that the probability of finding the drugs in the “lower” position (in the bottom of the cavity) is lower than in the “upper” one. In the case of compound **24**, the complexes of the drug with the “lower” site of the receptor have values ΔG > 0. It means that compound **24** cannot bind in the “lower” site at all. Compounds **13** and BIMU-8 bind to 5-HT_6_R at the same location as serotonin—at the bottom of the binding cavity (Figure 10B). Compound **24**, according to molecular docking data, does not interact with 5-HT_6_R. Finally, in the case of 5-HT_7_R, all three compounds bind to the receptor at the same site as the serotonin derivative 3-(2-azanylethyl)-1H-indole-5-carboxamide (8K3) (Figure 10C).

According to molecular docking data, benzimidazole–carboxamides bind to 5-HT_4_R most strongly among three types of 5-HTR. Experimental data on the effectiveness of the interaction of compounds **13** and **24** with 5-HTR have not yet been obtained. However, since these drugs are analogs of BIMU-8 (which is a specific agonist of 5-HT_4_R), it is quite expected that the efficiency is maximum for 5-HT_4_R. Compounds **13**, **24** and BIMU-8 interact less efficiently with 5-HT_7_R and very weakly (or do not interact) with 5-HT_6_R.

### 3.7. Interaction of Compounds ***13***, ***15*** and ***24*** with 5-HT_4_R According to MD Simulation

Using the method of MD in the lipid bilayer, we tested the stability of the complexes of 5-HT_4_R with compounds BIMU-8, **13** and **24**. This subtype of the serotonin receptor was chosen for two reasons: firstly, the studied drugs bind most efficiently to 5-HT_4_R, and secondly, the fact that these compounds and serotonin bind in different locations according to molecular docking data requires additional investigation. The complexes of the drugs with an upper site of 5-HT_4_R (Figure 10A) as well as “forced” complexes of BIMU-8 and compound **13** with a lower site (serotonin-binding location) of 5-HT_4_R served as the starting conformations for MD simulations.

At the current stage of our research, we were not interested in the conformational changes of associated G-protein caused by the binding of agonists to 5-HT_4_R. So, we simulated only the ligands and the receptor inserted into the lipid bilayer without G-protein included in the system (although the structure of the complex of 5-HT_4_R with G-protein is available in the PDB database). The preparation of the 3D models is described in the Methods section; Figure 11 shows the model of the complex of compound **13** with 5-HT_4_R in the lipid bilayer with a solvent (water) as an example.

Conformational changes of the complexes of BIMU-8 with the upper and lower binding sites of 5-HT_4_R are shown in Figure 12.

According to the obtained data, BIMU-8 comes to the same location within the 5-HT_4_R receptor both from the upper (Figure 12A) and lower (Figure 12B) positions. This resulting location we believe to be the site of interaction of BIMU-8 with the receptor. Apparently, this site was not revealed with molecular docking since BIMU-8 can only get there after binding to the surface of 5-HT_4_R in the upper site, after which conformational changes occur in the receptor opening the passage to this site.

Conformational changes of the complexes of compound **13** with the upper and lower binding sites of 5-HT_4_R are shown in Figure 13.

In the case of the upper site, during 100 ns of simulation, compound **13** is extended along the axis of the receptor, but does not move deeper (Figure 13A). However, compound **13** moves to the same site described above for BIMU-8 when starting from the serotonin-binding site (Figure 13B). It means that compound **13** might share the binding site with BIMU-8, but cannot reach it due to the massive tert-butyl group or more time needed to penetrate inside. Long MD simulation together with in vitro experiments will give a clearer answer, which is the task of our further studies.

Conformational changes of the complex of compound **24** with the upper binding sites of 5-HT_4_R are shown in Figure 14.

Compound **24** remains in the same location during the simulation, and only the position of the drug relative to the nearest amino acids is corrected (Figure 14). Therefore, compound **24** is either not an agonist of 5-HT_4_R at all or it works as a modulator of 5-HT_4_R activity binding in the upper site. Such modulators are described in the literature for the serotonin receptor 5-HT_2C_ (5-HT_2C_R) [63].

### 3.8. Interaction of Compounds ***22***, ***23*** and ***27*** with 5-HT_4_R According to Molecular Modeling Data

Additionally, using molecular modeling methods, we studied how the structure of substituents R1, R2 and R3 (Figure 1) affects the interaction of benzimidazolones with 5-HT_4_R. For this additional experiment, we selected compounds **22** (diethyl group in substituent R3), **23** (trifluoromethyl group in substituent R1) and **27** (cyclopropyl group in substituent R2 and diethyl group in substituent R3). According to molecular docking data, these compounds bind initially in the same location as BIMU-8 (at the entrance to the cavity of the binding site); the ΔG values are −7.2, −8.1 and −7.5 kcal/mol for compounds **22**, **23** and **27**, respectively. According to the MD simulation, only compound **22** moves deeper towards the bottom of the binding site cavity (Figure 15A) and after 100 ns of simulation, stops in the same location as BIMU-8 (Figure 15A,B). Compound **23** sinks only slightly into the cavity of the binding site (Figure 15B), while compound **27** extends along the axis of the receptor but does not move further towards the bottom of the cavity (Figure 15C).

## 4. Discussion

The study of the biochemical properties of benzimidazolones began in 1999, when they were first synthesized. One of the first compounds of this class was BIMU-8, which is an agonist of 5-HT_4_R capable of penetrating through blood–tissue barriers; it also shows affinity for 5-HT_3_R [38]. Studies of its properties have shown that it alleviates respiratory depression by activating the pre-Bötzinger complex located in the respiratory center of the brainstem. Also, the compound was found to be able to enhance brain activity and increase the ability in learning and memorizing [64]. It is believed that this is the result of a positive effect of activation of 5-HT_4_R on the production of acetylcholine [65].

Multifunctional ligands are supposed to be used not only for symptomatic but also for pathogenetic therapy. However, the implementation of this idea is associated with a number of interrelated problems: (a) a high molecular weight of hybrid molecules; (b) low oral bioavailability; (c) inability to cross the blood–brain barrier; (d) action of the components of the hybrid molecule in various concentration ranges; and (e) need to optimize selectivity and affinity for different targets. It is necessary to take into account these problems and make appropriate calculations at the very first stages before proceeding to animal studies. At the same time, it should be understood that the degree of agreement between the data obtained with calculation and experimental methods is a separate issue.

The algorithm of the approach that we proposed (described in [37] and in the present study) consists in the formation of a group of substituted 1,3-dihydro-2-oxo-1H-benzimidazol-2-ones at the first stage for subsequent synthesis. For this purpose, such characteristics as logP, logBB and LD_50_ (mg/kg) for mice and rats were calculated in silico [37]. Based on the data obtained, the most promising compounds were selected, which were subsequently synthesized, and their anticholinesterase activity was studied in vitro. And finally, using the methods of molecular modeling, we studied the interaction of the compounds with ChE and three types of G_s_-protein-coupled 5-HTR (5-HT_4_R, 5-HT_6_R and 5-HT_7_R). Then, based on a combination of in vitro and in silico data, we determined the structural features of the drugs that can affect their anticholinesterase and serotoninergic activity.

The biochemical analysis performed here showed the ability of the obtained structures to inhibit AChE and BChE, which expands the scope of their application as potential dual-action drugs (functional binary molecules). The compounds, in general, showed moderate inhibitory activity towards ChE and inhibited BChE more strongly than AChE (Appendix A). The introduction of the trifluoromethyl group into the benzene ring generally increased the anticholinesterase activity (compounds **15** and **16** as well as **22** and **23**). As for the effect of diethylaminoethyl and dimethylaminoethyl substituents in the carboxamide moiety, compound **27** containing a diethylaminoethyl substituent had improved inhibitory properties compared to compound **28** containing a dimethylaminoethyl one. However, in general, the activity of **27** and **28** was weaker than that of compounds **13**, **22** and **23**, in which the imidazole ring contains the tert-butyl or iso-propyl substituent rather than the cyclo-propyl one. Compound **13** is the strongest inhibitor of BChE (K_i_ = 1.7 µM) and exhibits a significantly lower inhibitory activity against AChE (K_i_ = 36.1 µM). Compound 24 is the strongest inhibitor of AChE (K_i_ = 0.13 μM), but did not demonstrate inhibitory properties against BChE. Compound **13,** with the most hydrophobic fragment at position R2 and the most mobile fragment with a less rigid skeleton at position R3, shows better inhibitory properties against BChE and lower toxicity [37] compared to BIMU-8.

According to molecular docking and molecular dynamics data, the site of competitive inhibitors S1a is the main site of interaction of compound **13** with AChE, which is consistent with our experimental data indicating that compound **13** is a competitive inhibitor of AChE. In the case of BChE, compound **13** interacts with approximately equal efficiency with both the site of competitive inhibitors S1b and allosteric Site S2b, which is inconsistent with our in vitro experiment. According to in silico data, compound **24** does not interact with Site S1a (apparently due to steric restrictions), but it effectively and rather strongly binds to allosteric Site S2a, which corresponds to the result of our biochemical experiment. Compound **24** binds to Site S1b of BChE with almost zero efficiency (and cannot compete with ACh for binding to the active site) and does not interact with Site S2b at all, which is again consistent with in vitro data on the absence of an inhibitory effect of compound **24** on BChE activity. When comparing the values of K_i_ obtained in vitro (Appendix A) with the calculated values of ΔG (Table 3), it can be noted that, for compound **13**, the effect of the “overestimation” of the efficiency of its interaction with the sites of competitive inhibitors of AChE and BChE takes place in computational experiments. Thus, according to the values of the constants shown in Appendix A, compound **13** inhibits AChE almost 300 times less effectively than compound **24** (K_i_ values are 36.1 and 0.13 μM, respectively), while the calculated values of ΔG differ by only 0.4 kcal/mol (−7.5 kcal/mol for **13** in Site S1a and −7.9 kcal/mol for **24** in Site S2a). Compound **13** is a strictly non-competitive inhibitor of BChE (Appendix A); however, according to molecular modeling, compound **13** interacts with both Sites S1b and S2b with approximately equal efficiency (even 0.2 kcal/mol is more efficient with Site S1b).

This effect of “overestimation” of efficiency can be explained by the fact that the interaction with the cavity of Sites S1a and S1b, located deep in the ChE globule, occurs gradually. At the first stage, a ligand anchors in PAS and then it gradually moves deep into the active site. Apparently, this additional (limiting) binding step, which is difficult to take into account when docking the ligand inside the active site, reduces the efficiency of the interaction of compound **13** with Sites S1a and S1b in vitro. The interaction of **13** with the allosteric centers of ChE can also have an additional effect, which can lead to some conformational changes in the active center and decrease the efficiency of the interaction of the ligand with Sites S1a and S1b. This effect is worthy of additional computational experiments, which will expand knowledge about the mechanisms of ChE inhibition. It is the task of our next study.

It is also interesting to note that, according to molecular docking data, compound **13** binds in one region on the surface of BChE (Site S2b, corresponds to a part of Site S2a on the surface of AChE when AChE and BChE molecules are superimposed), and, during MD simulation, it passes to the neighboring region (S2b′), which was not detected with blind docking (Figure 16).

When AChE and BChE are superimposed, Site S2b′ corresponds to another part of Site S2a of AChE. Thus, regions S2b and S2b′ can be considered as two chambers of one large allosteric site on the BChE surface. Probably, the difference between the binding modes of **13** inside BChE obtained with molecular docking and MD can be explained by the fact that binding in one chamber (S2b) of the allosteric site changes the surface configuration and lets the substrate move into another one (S2b′). The second identified allosteric site on the surface of AChE (S3a) cannot retain benzimidazolone molecules (Figure 7C and Figure 8B) and does not appear to be a binding site for this class of compounds.

On the basis of molecular modeling data on the interaction of compounds **13** and **24** with the ligand-binding sites of AChE and BChE, we can try to explain some experimental observations about the anticholinesterase activity of benzimidazole–carboxamides. Thus, according to the data shown in Appendix A, benzimidazole–carboxamides inhibit BChE more pronouncedly than AChE. This fact can partly be explained by the smaller steric size of the active site of AChE compared to BChE, which makes it more difficult for massive benzimidazole–carboxamides to get into the active center of AChE. According to MD simulation, when compound **13** is bound in Site S1a (the site of competitive inhibitors of AChE), its cationic group interacts with Tyr337, and the diethyl substituent of the cationic group binds even deeper in the active site gorge, closer to the catalytic triad (Figure 4A). Obviously, compound **24**, due to its massive cationic group (Figure 1), cannot pass into Site S1a.

It can also be noted that the competitive type of inhibition is characterized for BChE, while the mixed type is for AChE. In all likelihood, the explanation for this phenomenon also lies in the fact that the active center of BChE has a large steric size compared to AChE. The only exception in the case of AChE are compounds **13** and **22**, which are strictly competitive inhibitors of AChE. It means that these compounds cannot interact (or interact very weakly) with the allosteric site of AChE (S2a). We assume that this may be due to their steric size. On the one hand, the compounds are bulky enough (compared, for example, with compound **28**) to sink deeper to the bottom of S2a and fixate there. On the other hand, **13** and **22** are not so massive (compared, for example, with compounds **15**, **16** and **27)** as to occupy the entire space of the site and stay inside due to steric interactions. Moreover, **13** and **22** do not contain halogen substituents in their structure (like compounds **16**, **23** and **24**). The additional negative charge on the halogen substituents may contribute to faster drug binding in the AChE allosteric site. Thus, according to molecular docking data, the chlorine atom of compound **24** can form a halogen bond with the backbone oxygen atom of Pro537.

As for BChE, compounds **13**, **23** and **28** inhibit the enzyme in a non-competitive manner, while compounds **15**, **16**, **22** and **27** conduct it in a competitive manner. Compound **22** differs from **13** only in the absence of a third methyl group in substituent R2 (isopropyl in the case of **22** and tert-butyl in the case of **13**). It can be assumed that the additional methyl group of compound **13** makes it bulkier, so it is more difficult for it to get into the active site of BChE (S1b, the site of competitive inhibitors). According to MD simulation, in Site S1b, compound **13** is immersed into the site by rotating the benzimidazole group (and its substituents R1 and R2) deep into the site around the immobile cationic group anchored near Trp82 (Figure 9A). Therefore, compounds with a less massive R2 substituent can sink deeper into the site and be more strongly retained inside. On the other hand, the massive tert-butyl group helps compound **13** to bind more strongly to allosteric Site S2b compared to the compounds having the isopropyl or cyclopropyl group in the R2 substituent. According to MD simulation, the tert-butyl group of substance 13 binds in Site S2b surrounded by the hydrophobic groups of residues Tyr396, Pro401 and Phe526; the more hydrophobic substituent R2 is, the stronger the binding. Compound **28** is an exception to this trend because of its small size, due to which it can quickly sink to the bottom of the allosteric site of BChE, so it is more energetically beneficial for this compound to bind to Site S2b.

Compounds **16** and **23** (containing the trifluoromethyl group) have an interesting feature. They differ in substituent R3, the replacement of which changes the type of BChE inhibition from competitive to non-competitive. At the same time, the same substitution of R3 for a pair of compounds **15** and **22** (which do not contain the trifluoromethyl group) does not affect the type of inhibition. This paradox can be easily explained by referring to Figure 6A, which shows the position of compound **13** in the site of competitive inhibitors of BChE S1b. If the benzimidazole ring contained the negatively charged trifluoromethyl substituent (as in compound **23**), it would bind next to the negatively charged Asp70 of PAS, which is extremely unfavorable energetically. Therefore, compound **23** does not bind in Site S1b. In compound **16**, the cationic group (which binds near Trp82) is one chemical bond further from the benzimidazole ring, so in the case of substance **16**, the trifluoromethyl substituent would be located farther from PAS, and compound **16** can be a competitive inhibitor of BChE. In the case of compounds **15** and **22**, the absence of the trifluoromethyl group makes the length of the molecule not a critical parameter.

As for the interaction of the studied compounds with 5-HT_4_R, 5-HT_6_R and 5-HT_7_R, the information described in Section 3.6 and Section 3.7 gives only a preliminary evaluation of their serotonergic activity; however, it allows us to state that the drugs bind to 5-HT_4_R and 5-HT_7_R no weaker than to ChE, i.e., at therapeutically relevant concentrations. With the help of MD simulation, we have identified the mechanism of binding of the benzimidazole–carboxamides to 5-HT_4_R. With the example of BIMU-8, this mechanism is shown in Figure 17.

First, the compound binds on the surface of the receptor, then passes to the internal binding site. However, due to its steric size, BIMU-8 cannot sink to the very bottom of the cavity like serotonin does (Figure 17). The narrowest point of the corridor leading to the inner binding site can be seen in Figure 17. This narrow spot can make it difficult for BIMU-8 analogs with more massive substituents R1 and R2 (such as compounds **13** and **23**) to pass inside. It is interesting to note that compound **27** with cyclopropyl substituent R2 (smaller than the isopropyl substituent in BIMU-8) also does not reach the internal binding site within 100 ns of the simulation. We think that this result can be explained by the rigidity and lower conformational mobility of the cyclopropyl substituent.

It should be noted that the strong binding of compounds to 5-HT_4_R, 5-HT_6_R and 5-HT_7_R does not yet mean that it will lead to the proper conformational changes in the receptor molecule and further signal transmission. The study of the interaction of benzimidazole–carboxamides with 5-HTR in vitro, as well as a detailed conformational analysis of drug complexes with the receptors and associated G-proteins, are the goals of our future research. Nevertheless, in conclusion, we allow ourselves a little speculation regarding the ratio of calculated and experimental values. The experimentally obtained K_i_ values of compound **13** against AChE and BChE are 36.1 and 1.7 μM, respectively, and the calculated values of ΔG are −7.5 and −6.6 kcal/mol for Sites S1a of AChE and S2b of BChE. Thus, there is a clear discrepancy between the theoretical and experimental values, so that the question arises about the validity of the calculated values of the affinity of the substance for receptors, obtained in this work exclusively with the docking method. We explained above that the “overestimation” of the efficiency of binding compound **13** to the S1a sites of AChE may be due to the fact that the docking method does not take into account the primary sorption of compound **13** in PAS, so we suggest focusing on the following ratio: the value of ΔG of −7 kcal/mol obtained in an in silico experiment corresponds to an experimental K_i_ value of about 1 μM. In the case of serotonin receptors, the EC_50_ value for compound **15** (BIMU-8) relative to 5-HT_4_R is known to be 18 nM [66]. Estimated values of ΔG for complexes of compounds **13** and **15** are −7.8 and −8.5 kcal/mol, respectively. Thus, comparing all known data, we believe that the value of EC_50_ in relation to 5-HT_4_R for compound **13** can lie in the range of tens to hundreds of nM, and for 5-HT_7_R, it is an order of magnitude higher, in the range of hundreds to thousands of nM.

According to our data published in [37], compound **13** is one of the least toxic. The fact that compound **13** interacts with 5-HTR less efficiently than BIMU-8 indicates a high chance of using therapeutically relevant concentrations of compound **13** for both 5-HTR agonism and BChE inhibition.

## 5. Conclusions

According to the results of our studies presented in [37] and here, compound **13** has minimal oral toxicity and high inhibitory ability against BChE, along with low inhibitory ability against AChE. The primary in silico evaluation of serotoninergic activity showed that compound **13** interacts with 5-HT_4_R by a mechanism similar to the known selective agonist BIMU-8; however, it is possible that due to the massive tert-butyl group, compound **13** is a less effective 5-HT_4_R agonist than BIMU-8. Additionally, according to molecular modeling data, compound **13** effectively interacts with the serotonin-binding center of 5-HT_7_R. The pool of the obtained data allows us to mark compound **13** as the most promising for further experimental investigation (in vitro experiments with 5-HT_4,7_R and then, in the case of success, in vivo testing in animal models of neurodegenerative diseases), which will confirm or disprove the potential of compound **13** as a drug for AD management.

## Figures and Tables

**Figure 1 pharmaceutics-15-02159-f001:**
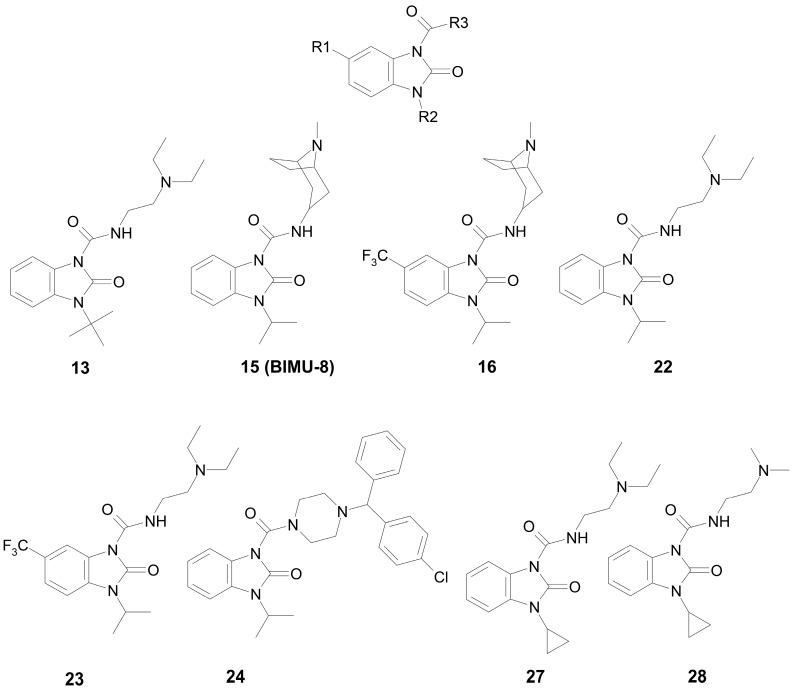
The structures of the target benzimidazole–carboxamides (in the canonical neutral form).

**Figure 2 pharmaceutics-15-02159-f002:**
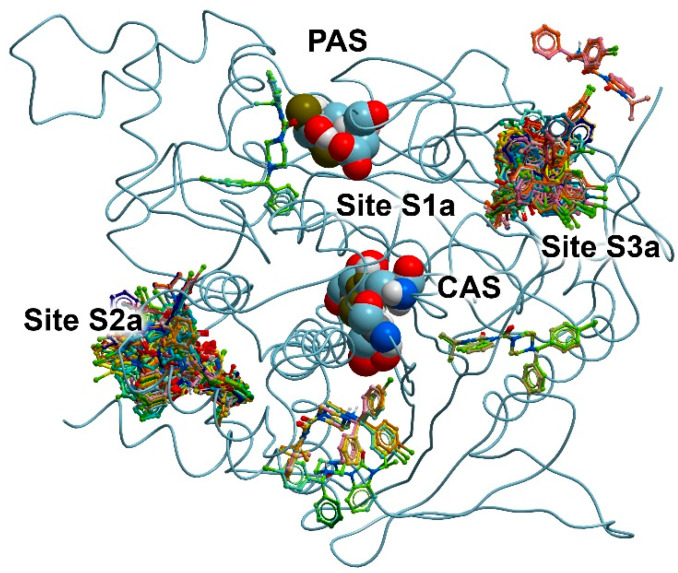
Binding sites for compound **24** on the surface of AChE according to blind molecular docking. The found positions of compound **24** are shown as balls and sticks. The catalytic active site (CAS, Ser203, His447 and Glu334) and the peripheral anionic site (PAS, Tyr72 and Asp74) of AChE are shown as spheres. The space between CAS and PAS forms Site 1a (S1a); allosteric sites on the surface of AChE are designated as Site S2a and Site S3a.

**Figure 3 pharmaceutics-15-02159-f003:**
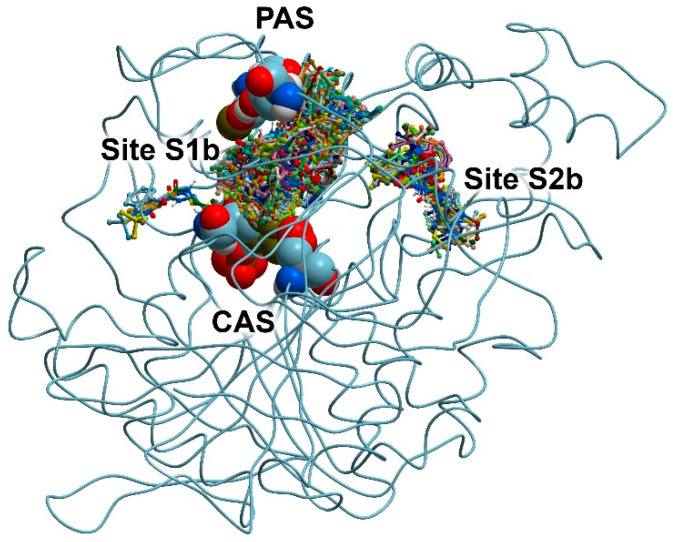
Binding sites for compound **13** on the surface of BChE according to blind molecular docking. The found positions of compound **13** are shown as balls and sticks. CAS (Ser198, His438 and Glu325) and PAS (Asp70 and Tyr332) are shown using spheres. The space between CAS and PAS forms Site S1b (the site of competitive inhibitors); the allosteric center on the BChE surface is designated as Site S2b.

**Figure 4 pharmaceutics-15-02159-f004:**
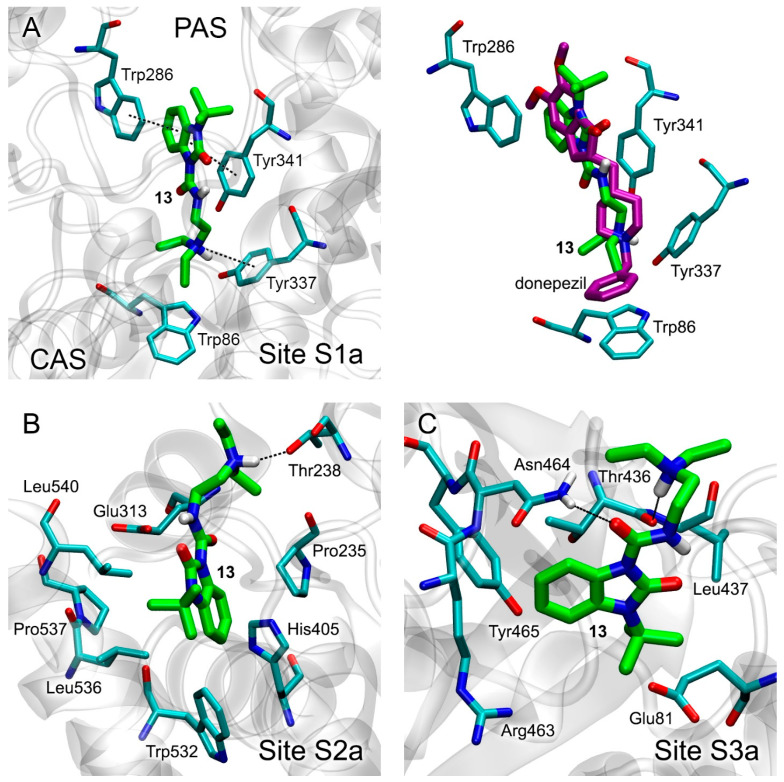
Binding modes of compound **13** in AChE ligand-binding sites S1a (**A**), S2a (**B**) and S3a (**C**) according to molecular docking data. The right panel of Figure 4A shows the alignment of positions of compound **13** (obtained with docking) and the donepezil molecule in Site S1a (obtained with an X-ray; PDB id 7e3h [59]). The carbon atoms of compound 13 are shown in green. The carbon atoms of donepezil are shown in magenta. Key interactions are shown with dotted lines. Non-essential hydrogens are omitted for clarity. CAS, catalytic active site; PAS, peripheral anionic site.

**Figure 5 pharmaceutics-15-02159-f005:**
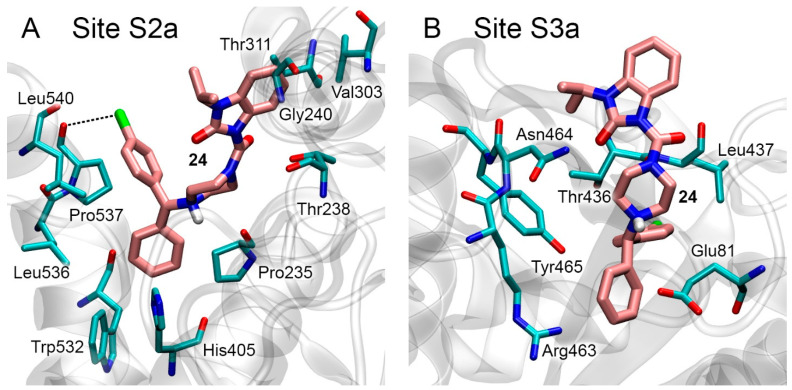
Binding modes of compound **24** in AChE allosteric sites S2a (**A**) and S3a (**B**) according to molecular docking data. The carbon atoms of compound **24** are shown in pink. Key interactions are shown with dotted lines. Non-essential hydrogens are omitted for clarity.

**Figure 6 pharmaceutics-15-02159-f006:**
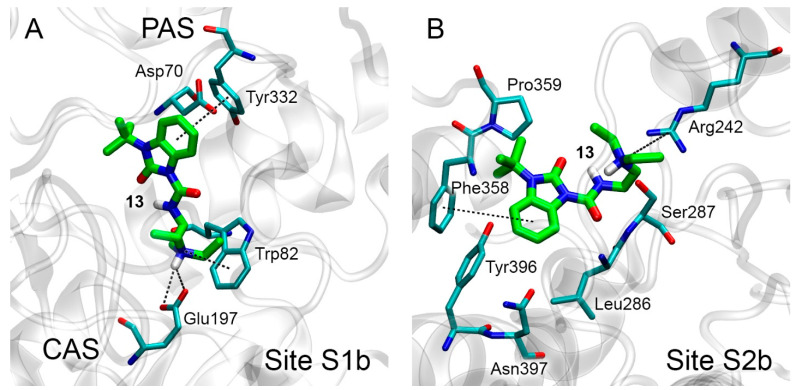
Binding modes of compound **13** in BChE binding sites S1b (**A**) and S2b (**B**) according to molecular docking. The carbon atoms of compound **13** are shown in green. Key interactions are shown with dotted lines. Non-essential hydrogens are omitted for clarity.

**Figure 7 pharmaceutics-15-02159-f007:**
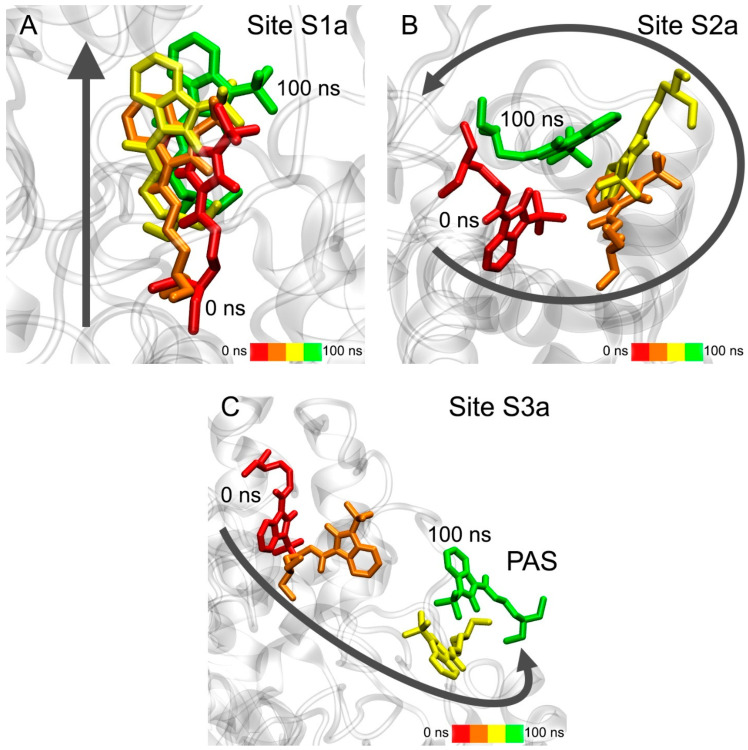
Interaction of compound **13** with AChE ligand-binding sites according to MD simulation. (**A**) Site S1a; (**B**) Site S2a; (**C**) Site S3a. CAS, catalytic active site; PAS, peripheral anionic site.

**Figure 8 pharmaceutics-15-02159-f008:**
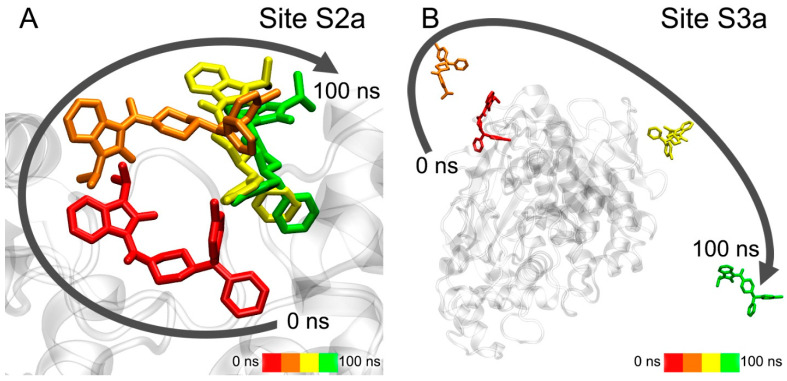
Interaction of compound **24** with AChE allosteric sites according to MD simulation. (**A**) Site S2a; (**B**) Site S3a.

**Figure 9 pharmaceutics-15-02159-f009:**
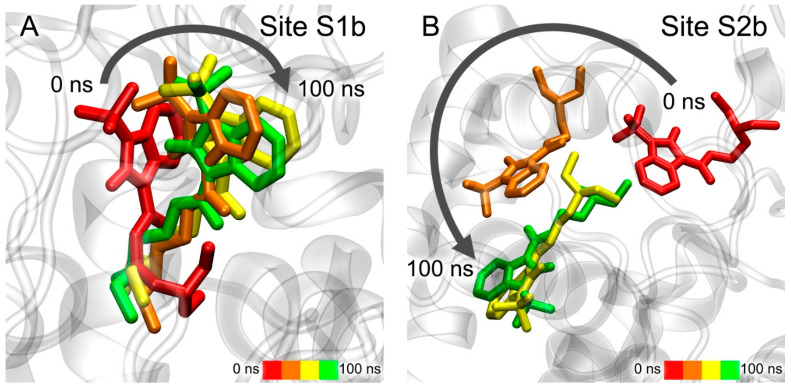
Interaction of compound **13** with BChE ligand-binding sites according to MD simulation. (**A**) Site S1b (site of competitive inhibitors); (**B**) Site S2b.

**Figure 10 pharmaceutics-15-02159-f010:**
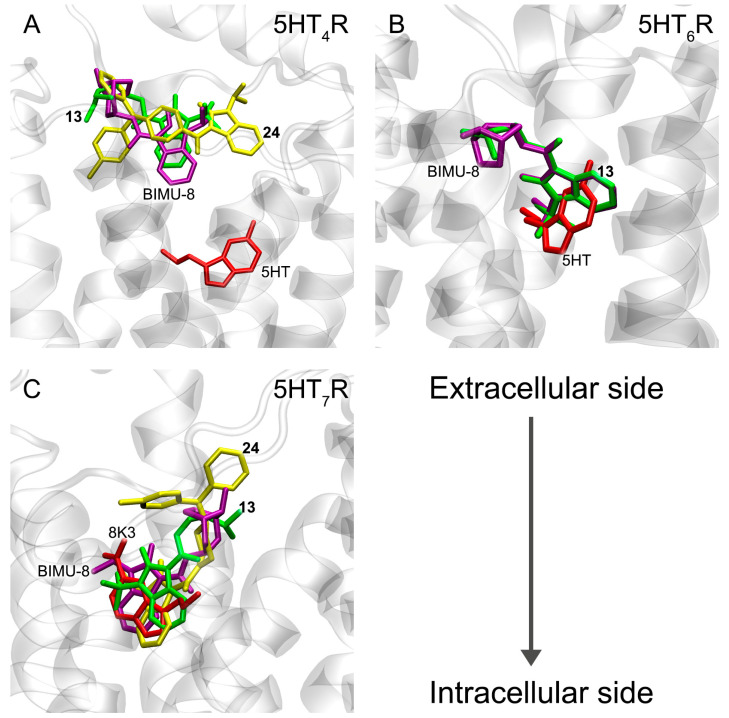
Binding modes of compounds **13** (green), **24** (yellow) and BIMU-8 (magenta) in the binding sites of 5-HT_4_R (**A**), 5-HT_6_R (**B**) and 5-HT_7_R (**C**) according to molecular docking data. The red color shows the binding modes of 5-HTR agonists obtained with electron microscopy [44]; serotonin (5HT) in the case of 5-HT_4_R and 5-HT_6_R or 3-(2-azanylethyl)-1H-indole-5-carboxamide (8K3) in the case of 5-HT_7_R. The arrow shows the orientation of the complexes relative to the extracellular and intracellular sides of the plasmatic membrane.

**Figure 11 pharmaceutics-15-02159-f011:**
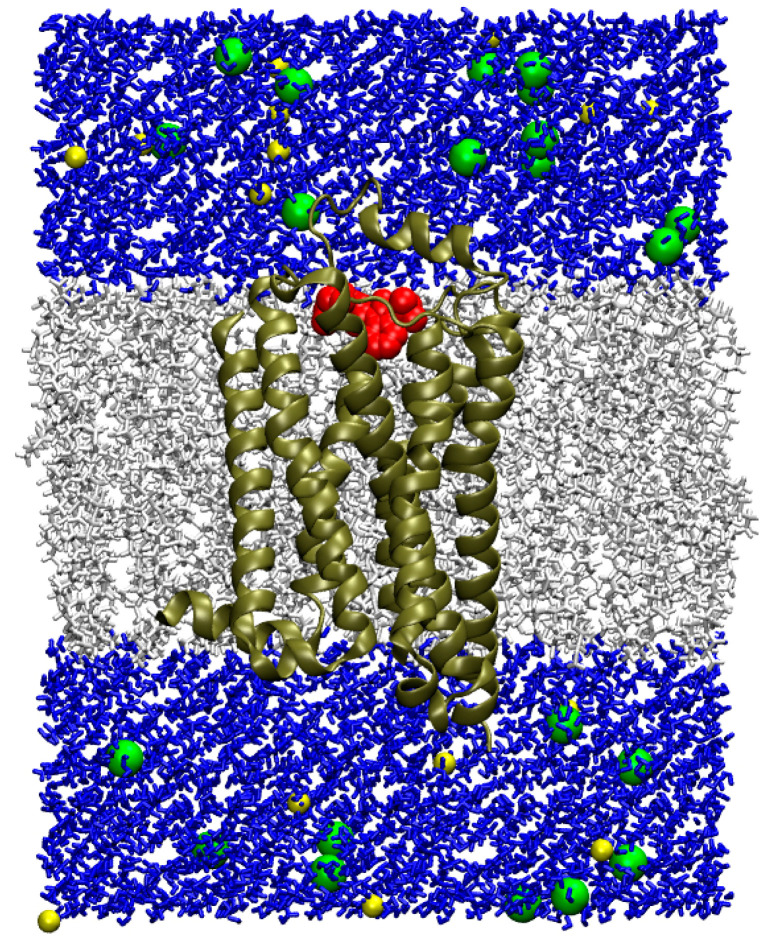
Three-dimensional model of the complex of compound **13** with 5-HT_4_R in the lipid bilayer. The receptor is shown as a brown ribbon, compound **13** as red spheres, the lipid bilayer as gray sticks, water as blue sticks and sodium and chloride ions as yellow and green spheres, respectively.

**Figure 12 pharmaceutics-15-02159-f012:**
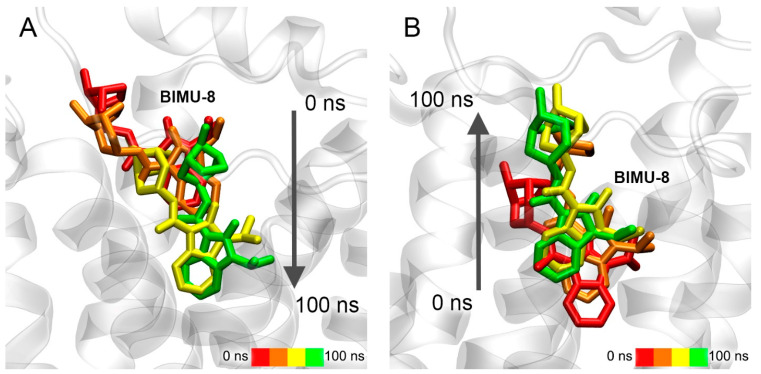
Interaction of BIMU-8 with 5-HT_4_R according to MD simulation. (**A**) movement of BIMU-8 after binding to the surface of the receptor (in the upper binding site); (**B**) movement of BIMU-8 out of the serotonin-binding site.

**Figure 13 pharmaceutics-15-02159-f013:**
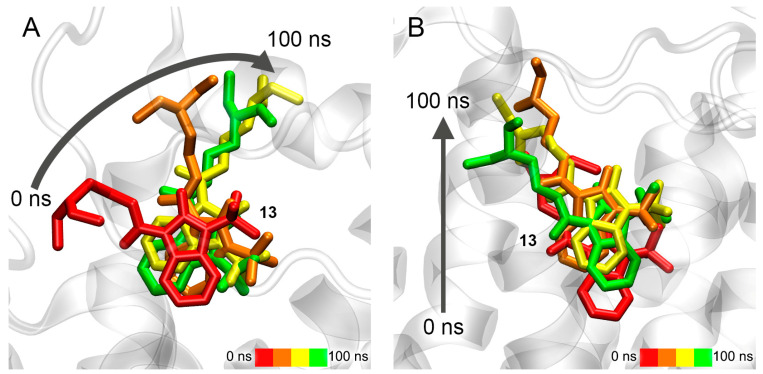
Interaction of compound **13** with 5-HT_4_R according to MD simulation. (**A**) movement of compound **13** after binding to the surface of the receptor (in the upper binding site); (**B**) movement of compound **13** out of the serotonin-binding site.

**Figure 14 pharmaceutics-15-02159-f014:**
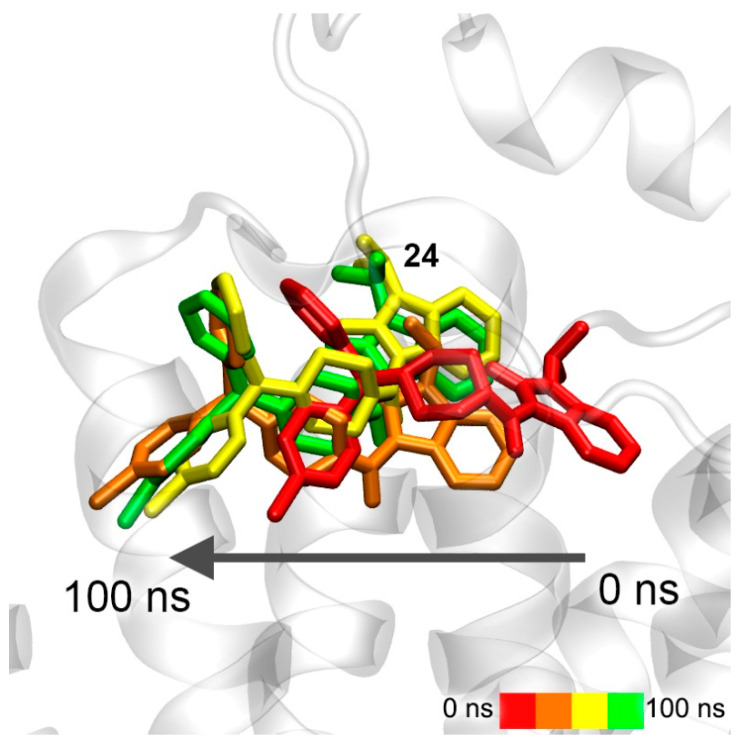
Interaction of compound **24** with 5-HT_4_R according to MD simulation.

**Figure 15 pharmaceutics-15-02159-f015:**
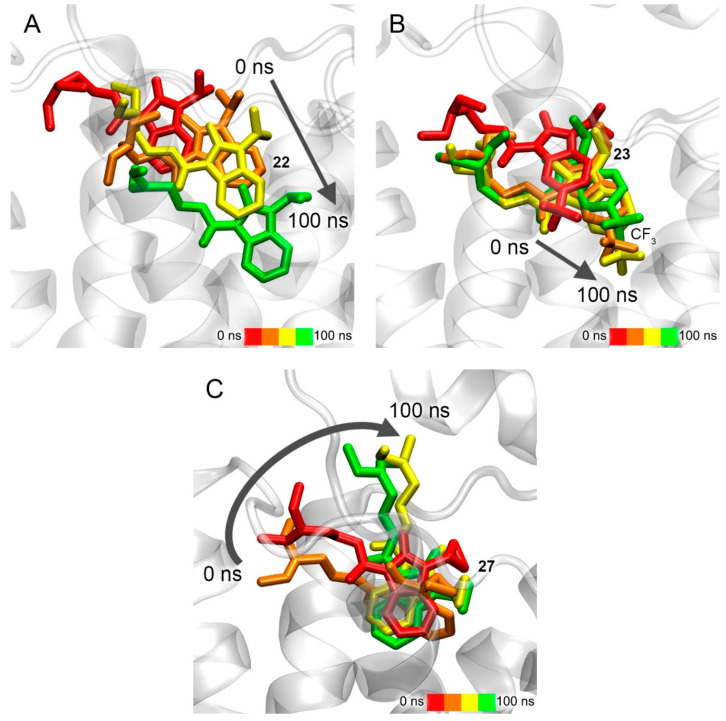
Interaction of compounds **22** (**A**), **23** (**B**) and **27** (**C**) with 5-HT_4_R according to MD simulation.

**Figure 16 pharmaceutics-15-02159-f016:**
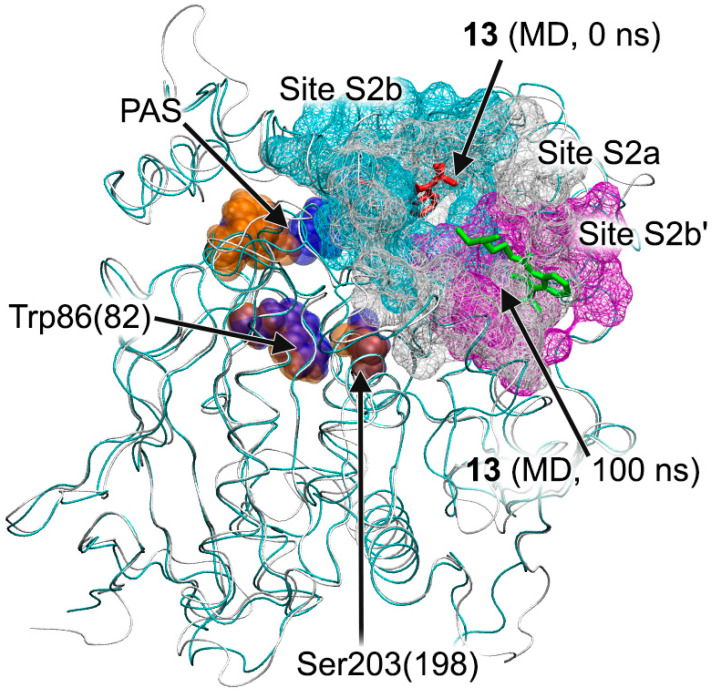
Topology of allosteric sites S2a and S2b on the surface of AChE and BChE. Molecules of AChE (grey ribbon) and BChE (blue ribbon) are superimposed. Site S2a of AChE (marked in gray), Site S2b of BChE (site for compound **13** according to molecular docking, marked in blue) and Site S2b′ (site for compound **13** according to MD, marked in magenta) are shown with a wireframe surface. The starting (docked) position of compound **13** is marked with red sticks, the final one after 100 ns of MD simulation is marked with green sticks. The key amino acids of the active site of AChE (orange) and BChE (blue) are marked with spheres: PAS, Trp86(82) and Ser203(198) (numbering is given for AChE and in brackets for BChE).

**Figure 17 pharmaceutics-15-02159-f017:**
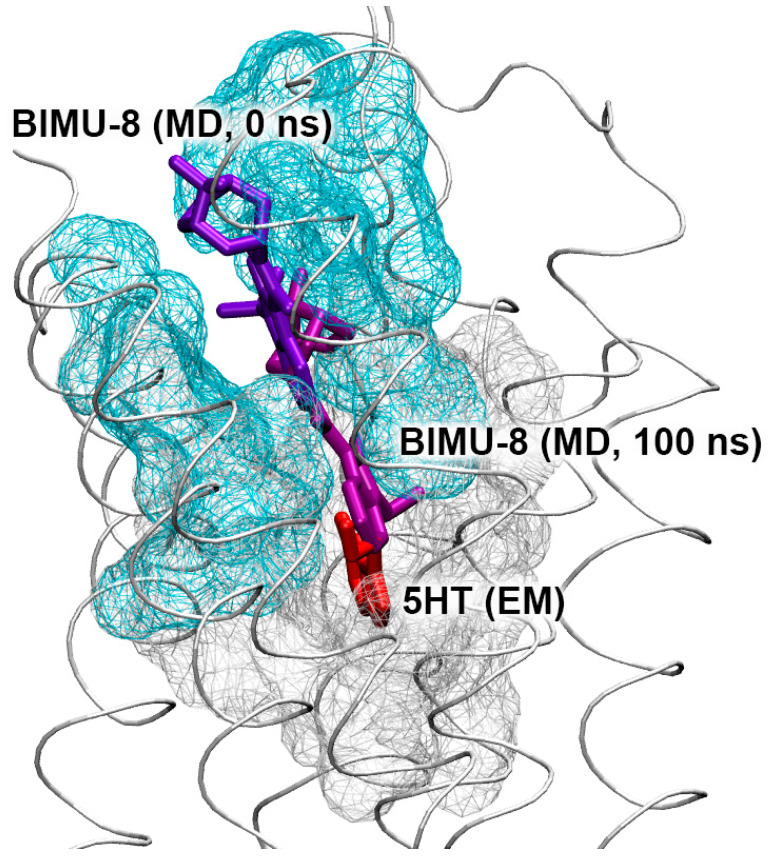
Topology of the ligand-binding site of 5-HT_4_R. The receptor is represented by the gray ribbon. The peripheral (primary) and internal binding sites are shown as blue and gray wireframe surfaces, respectively. BIMU-8 conformations at the peripheral and internal binding sites according to molecular docking and MD simulation are presented using violet and purple sticks, respectively. The conformation of serotonin (5HT) in the internal binding site according to electron microscopy (EM; PDB ID 7xta [44]) is shown using red sticks.

**Table 1 pharmaceutics-15-02159-t001:** Inhibitory activity of benzimidazole–carboxamides against human AChE.

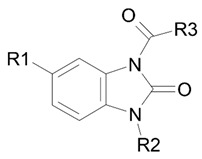					
Compound	R1	R2	R3	Ki, μM	Type of Inhibition
**13**	H	*t*-Bu	N,N-(Diethylamino)ethyl	36.1	competitive
**15** (BIMU-8)	H	*i*-Pr	8-methyl-8-azabicyclo[3.2.1]octan-6-yl	126.9	mixed (α = 6.6)
**16**	CF_3_	*i*-Pr	8-methyl-8-azabicyclo[3.2.1]octan-6-yl	31.0	mixed (α = 5.1)
**22**	H	*i*-Pr	N,N-(diethylamino)ethyl	97.1	competitive
**23**	CF_3_	*i*-Pr	N,N-(diethylamino)ethyl	60.5	mixed (α = 3.1)
**24**	H	*i*-Pr	[(4-chlorophenyl)(phenyl)methyl]-piperazin-1-yl	0.13	non-competitive
**27**	H	*c*-Pr	N,N-(diethylamino)ethyl	131.8	mixed (α = 10.9)
**28**	H	*c*-Pr	N,N-(dimethylamino)ethyl	137.9	mixed (α = 12.6)

**Table 2 pharmaceutics-15-02159-t002:** Inhibitory activity of benzimidazole–carboxamides against human BChE.

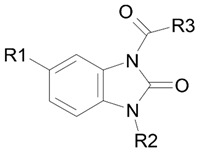					
Compound	R1	R2	R3	Ki, μM	Type of Inhibition
**13**	H	*t*-Bu	N,N-(Diethylamino)ethyl	1.7	non-competitive
**15** (BIMU-8)	H	*i*-Pr	8-methyl-8-azabicyclo[3.2.1]octan-6-yl	9.7	competitive
**16**	CF_3_	*i*-Pr	8-methyl-8-azabicyclo[3.2.1]octan-6-yl	21.3	competitive
**22**	H	*i*-Pr	N,N-(diethylamino)ethyl	15.2	competitive
**23**	CF_3_	*i*-Pr	N,N-(diethylamino)ethyl	4.5	non-competitive
**24**	H	*i*-Pr	[(4-chlorophenyl)(phenyl)methyl]-piperazin-1-yl	no inhibition	
**27**	H	*c*-Pr	N,N-(diethylamino)ethyl	16.2	competitive
28	H	*c*-Pr	N,N-(dimethylamino)ethyl	28.1	non-competitive

**Table 3 pharmaceutics-15-02159-t003:** Free binding energies (ΔG) of the complexes of benzimidazole–carboxamides with AChE and BChE binding sites according to molecular docking data.

Compound	AChE			BChE	
	S1a	S2a	S3a	S1b	S2b
**13**	−7.5	−6.7	−5.7	−6.8	−6.6
**24**	No Int	−7.9	−7.4	−0.3	No Int
ACh	−4.7			−4.3	

No Int, no interaction (ΔG > 0).

**Table 4 pharmaceutics-15-02159-t004:** Free binding energies (ΔG, kcal/mol) of the complexes of benzimidazole–carboxamides with serotonin receptors 5-HT_4_, 5-HT_6_ and 5-HT_7_ (5-HT_4_R, 5-HT_6_R and 5-HT_7_R) according to molecular docking data.

Compound	5-HT_4_R	5-HT_6_R	5-HT_7_R
**13**	−7.8	−5.2	−7.5
**24**	−10.7	No Int	−8.9
**15** (BIMU-8)	−8.5	−4.6	−8.4

No Int—no interaction (ΔG > 0).

## Data Availability

The data presented in this study are available on request.

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
