# Peer review of "Anticholinesterase and Serotoninergic Evaluation of Benzimidazole–Carboxamides as Potential Multifunctional Agents for the Treatment of Alzheimer’s Disease"

_pharmaceutics, 2023, doi:10.3390/pharmaceutics15082159_

Round 1

Reviewer 1 Report

The research study entitled “Anticholinesterase and Serotoninergic Evaluation of Benzamide- 2

azole-Carboxamides as Potential Multifunctional Agents for the Treatment of Alzheimer's Disease”

by Belinskaia D.A et all have investigated the interaction of benzimidazol-2-ones with cholinesterase and serotonin receptors, which are potentially involved in the pathophysiology of AD. Molecular docking and molecular dynamics (MD) simulation in water and lipid bilayer were used to study the interaction of the compounds with ChE and 5-HTR. To confirm their hypotheses, they performed In vitro and in silico studies, the in vitro findings suggested that the said compounds had anticholinesterase activity, while the molecular modeling, the mechanism of interaction of the tested compounds with the ChE was investigated. Primary in silico evaluation showed that benzimidazole-carboxamides effectively bind to serotonin receptors 5-HTR4 and 5-HTR7. The data has finally suggested that compound 13 is the most effective drug against the two give factors (ChE and Serotoninergic in AD).

Overall, the study may provide the background for further studies related to in vivo and the mechanistic studies may be conducted. I haven’t checked the overall English style, and the methods of in silico as that’s not my expertise.

Overall, the paper is quite interesting, but I will add a few other suggestions to improve its quality.

1.       The authors have targeted the ChE and Serotonergic pathways, but the rationale for targeting these pathways is not satisfactory. There are some main cardinal features of AD, which may be targeted instead of serotonergic and cholinergic system.

2.       The graphical representation of the in vitro results in figure 2 is not a good representation of the findings. I will suggest changing it into an easily understandable for a common science student.

3.       The paper has given in vitro findings first and after that they have given them in silico, why not they reverse it.

4.       The current in vitro findings may not endorse what the authors have claimed, to confirm the hypothesis the authors may provide more in vitro results.

5.       I would like to see the revised version of the current manuscript.

The editorial member may consider it for publication, if the English style is improved.

 English is fine Minor editing  required

Reviewer 2 Report

The papaer of Belinskaia et al. describes thethe acetilcholinesterase and serotoninergic activity  of new benzimidazolon derivatives by docking experiments and acetylcholinesterase activity assay. The results are not breakthrought, they choosed compound 13 to most be effective.

There are several mistakes can be improved:

row 92: fluoride atom in the structure (give it more precisely)

row 502: drugs are analogues of BIMU

rows 697, 710 steric size? bulk size?

Compounds in the SM are not properly characterized (IR, MS 13C-NMR) melting point, color ect. are  missing.

Reviewer 3 Report

This manuscript describes several benzimidazole-carboxamide compounds as potential multifunctional drugs for the treatment of Alzheimer's disease. It has several shortcomings:

1.     The text on lines 33-38 is of very dubious validity and needs to be corrected/deleted.

2.     The rationale combining cholinergic serotonergic activity in a single molecule as a treatment strategy for Alzheimer’s disease has not been made.

3.     The title refers to combining cholinesterase inhibition with serotonergic activity, yet the text refers to combining cholinesterase inhibition with GPCR activity. This is very confusing.

4.     The manuscript frequently refers to serotonin receptors, but fails to describe which subtypes of these receptors they are referring to. This is not acceptable.

fine

Round 2

Reviewer 2 Report

The authors corrected their manuscript as required, so the manuscript can be accepted.

Author Response

Since Reviewer #2 has no more questions or comments per se, we would like to thank him/her for appreciating our work.

Reviewer 3 Report

The Ki for compound 13 for AChE and BChE was shown as 36.1 and 1.7 micromolar, respectively. How does this compare with that for 5-HT4, 5-HT6 and 5-HT7 receptors?

ok

Author Response

Dear Reviewer,

Thank you so much for this additional question, our reply/explanation/speculation is in the file attached, and also additional fragment and reference highlighted green are inserted at the end of the manuscript.  
